# Cervical Sagittal Balance: Impact on Clinical Outcomes and Subsidence in Anterior Cervical Discectomy and Fusion

**DOI:** 10.3390/biomedicines11123310

**Published:** 2023-12-14

**Authors:** Adam Bębenek, Maciej Dominiak, Bartosz Godlewski

**Affiliations:** Department of Orthopaedics and Traumatology, with Spinal Surgery Ward, Scanmed—St. Raphael Hospital, 30-693 Cracow, Poland; bebenek.adam.24@gmail.com (A.B.); maciej.dominiak92@gmail.com (M.D.)

**Keywords:** ACDF, subsidence, sagittal balance

## Abstract

Degenerative disease of the cervical spine leads to sagittal imbalance, which may affect treatment results. The purpose of this study was to evaluate changes in selected cervical sagittal balance parameters and their effects on subsidence and clinical outcomes of the procedure. This study encompassed a total of 95 evaluated patients who underwent anterior cervical discectomy and fusion (ACDF). Selected cervical sagittal balance parameters were assessed using lateral projection X-rays: C2–C7 spinal vertical axis (C2–C7 SVA), spinocranial angle (SCA), C7 slope, C2–C7 lordosis, and the segmental Cobb angle. Measurements were collected the day before, the day after, and 12 months after surgery. Changes in clinical parameters was assessed using the VAS and NDI scales. Subsidence was defined as a loss of intervertebral height of more than 30% of the baseline value. Among all the assessed parameters, only the C2–C7 SVA demonstrated a statistically significant difference between the groups with and without subsidence: 26.03 vs. 21.79 [mm], with *p* = 0.0182, preoperatively and 27.80 vs. 24.94 [mm], with *p* = 0.0449, on the day after surgery, respectively. We conclude that higher preoperative and postoperative C2–C7 SVA values might contribute to an elevated risk of implant subsidence. Furthermore, both the SCA and C7 slope could conceivably influence the clinical outcome, respectively impacting pain, as assessed by the VAS and the disability, as evaluated through the NDI scale.

## 1. Introduction

Degenerative spinal disease not only leads to symptoms related to nerve structure irritation within the spinal canal and intervertebral foramina, but also contributes to the restricted mobility of specific spinal segments [1]. This compromised mobility results in the inability to effectively counterbalance the abnormal positioning of the body’s center of gravity. Consequently, the deviation of the body’s center of gravity from its usual axis prompts compensatory postural adjustments to restore the disrupted sagittal balance [1,2]. Nevertheless, this process operates within a detrimental cycle. Triggered by compensatory adjustments, peripheral joints become susceptible to accelerated degeneration. This, in turn, curtails their mobility, thus leading to the transmission of excess strain to the spine. Consequently, this exacerbates the advancement of spinal degeneration and amplifies the existing imbalance [1,2,3]. The described phenomena inevitably manifest in the cervical spine, which, due to its heightened mobility, is anticipated to counterbalance the disorder and uphold a horizontal gaze [4,5]. Central to the progression of these degenerative changes seems to be the occurrence of hyperlordosis, which causes simultaneous overloading of the intervertebral joints and the intervertebral disc [1]. In 2018, Ling et al. conducted a comprehensive literature review aimed at identifying the optimal parameters for evaluating sagittal balance in the cervical region [5]. While commonly employed metrics comprise the C2–C7 lordosis, C0–C2 lordosis, and the global cervical angle (Harrison method), Ling et al. highlighted the cervical sagittal vertical axis (cSVA), spinocranial angle (SCA), T1 slope/C7 slope, and C2–C7 lordosis as most valuable, owing to their correlation with spinopelvic balance [5,6]. Alterations in cervical sagittal balance (CSB) parameters have been investigated in both asymptomatic populations and instances necessitating surgical interventions for lesions; however, the latter group has been characterized by a more limited number of reports [4,7,8,9,10]. Anterior cervical discectomy and fusion (ACDF), the established gold standard treatment for cervical myelopathy, continues to be the prevailing procedure employed for cervical degenerative disease [11]. The procedure involves spinal canal decompression followed by the precise implantation of an intervertebral device. Careful selection of the implant, tailored to the disc space, offers the potential to adjust cervical sagittal balance, thereby influencing the broader spinopelvic sagittal balance to some extent [5,7]. One of the complications associated with ACDF, related to the implantation of the mentioned cage, is the phenomenon of subsidence, which involves the settling of the implant into the fused vertebral bodies [12,13]. The causes of this phenomenon might be attributed to bone density disorders or other risk factors that interfere with the osteointegration [12,14]. Studies have been undertaken to evaluate this process based on the relationship between the implant materials and the bone density, thereby assessing subsidence incidence and fusion rates [15,16]. Translational research based on nuanced laboratory, biomechanical, and radiological data appears to be necessary, as the subsidence phenomenon is still regarded as clinically inconclusive; however, there are reports suggesting that it can deteriorate treatment outcomes and impact balance [12,17,18,19]. The objective of this study was to assess how alterations in CSB parameters influence the clinical outcomes of the procedure measured by the VAS (visual analogue scale) and NDI (neck disability index), as well as the incidence of subsidence during a 12-month follow-up period.

## 2. Materials and Methods

### 2.1. Study Design

A single-center, observational study was conducted with 104 patients operated on at the authors’ center between 2019–2021 for cervical disc disease. Inclusion criteria were as follows: diagnosed cervical degenerative disc disease on preoperative MRI that did not respond to conservative treatment, age ≥ 18, and eligibility for single- or double-level ACDF surgery. Exclusion criteria included ages younger than 18 years, comorbidities that disqualified the patients from surgery, diagnosed osteoporosis, and those requiring three or more levels of surgery. During the study period, a total of 193 individuals were evaluated, of which 104 met the study criteria. As 9 patients were lost during follow-up, the images from the remaining 95 were assessed (Figure 1). The subjects’ mean age was 51, with a median of 50. The youngest participant was 31, while the eldest was 71. Among the participants, 67 (71%) were women (Table 1).

### 2.2. Procedure and Implants

All procedures were performed with the same surgical technique from the Smith–Robbinson approach [20]. All implanted intervertebral devices had the same length (11.5 mm) and width (14 mm), thus giving them the same surface area. They differed only in height and the material from which they were made. PEEK (polyether-ether-ketone,) and TC PEEK (titanium-coated PEEK) implants from a single manufacturer (Aesculap Chifa, CeSPACE^®^ Implants, Tuttlingen, Germany), were used, with a range of possible heights of 4–8 mm. Each implant was filled with nanoparticle hydroxyapatite from the same manufacturer (B Braun, Nanogel^®^ Hydroxyapatite, Melsungen, Germany). We used only stand-alone cages, without plating.

### 2.3. Radiological Assessment and Subsidence Criteria

Radiological parameters were assessed using X-rays in lateral projection at five time points: (1) the day before the surgery, (2) the day after the surgery, (3) one month after the surgery, (4) six months after the surgery, and (5) twelve months after the surgery. All radiographs were obtained at the authors’ center always using the same equipment and following the same procedure. Measurements were collected with an accuracy of one decimal place. The C2–C7 sagittal vertical axis (cSVA), spinocranial angle (SCA), C7 slope, C2–C7 lordosis, and segmental angle (Cobb) parameters were chosen to assess sagittal balance (see Table 2 and Figure 2). These parameters were selected based on a comprehensive systematic review conducted by Ling et al. in 2018, which recognized them as the most effective and dependable for evaluating CSB [5]. The established values for the measured parameters were those provided by Ling and Le Huec [2,5]. The criterion for subsidence was defined as the depression of the implant into the border plate by at least one-third of the intervertebral space’s height (Figure 3) [14,21].

### 2.4. Clinical Assessment

On the days when follow-up images were taken, clinical outcome was assessed using visual analogue scale (VAS) and neck disability index (NDI) scales [22,23]. A neck disability index (NDI) score of <15 points was considered indicative of mild disability with minimal interference in daily activities. Similarly, a visual analog scale (VAS) score of <1 point implied the absence of pain requiring medication. 

### 2.5. Statistical Analysis

The comparison of quantitative variables between the groups was performed using either the Mann–Whitney test or the Student’s *t* test for independent variables. To perform a multivariate evaluation of the effect of selected radiological parameters on subsidence, we utilized statistical analysis through logistic regression using the Wald test. A significance level of 0.05 was adopted for the analysis, thereby considering p values below 0.05 as indicating significant relationships. In instances where there were discrepancies in the statistical significance between univariate and multivariate tests, we prioritized the results of the multivariate tests. All calculations were carried out using MedCalc^®^ statistical software version 20.104 and TIBCO Statistica^®^ 13.3.

### 2.6. Ethical Approval 

The research was approved by the Bioethics Committee of the Andrzej Frycz Modrzewski University in Cracow (Resolution 4/2019) and was conducted in compliance with the Declaration of Helsinki. All qualified patients gave written consent to participate in the study.

## 3. Results

### 3.1. CSB Parameters

The preoperative measurements of the sagittal balance parameters within the cervical segment were recorded as follows: for the C2–C7 sagittal vertical axis (SVA), a mean of 23.5 mm (SD ± 11.5 mm) and a median of 22 mm were recorded; for the spinocranial angle (SCA), a mean of 81° (SD ± 9.8°) and a median of 79.8° were recorded; for the C2–C7 lordosis, a mean of 9.5° (SD ± 10.7°) and a median of 8.6° were recorded; for the C7 slope, a mean of 20.5° (SD ± 7.7°) and a median of 21° were recorded; and for the segmental Cobb angle, a mean of 6.2° (SD ± 6.1°) and a median of 5.5° were recorded. The postoperative sagittal balance parameters, measured from images taken the day after surgery, exhibited the following values: for the C2–C7 SVA, a mean of 26 mm (SD ± 10.6 mm) and a median of 24.4 mm were recorded; for the SCA, a mean of 79.5° (SD ± 7.3°) and a median of 79.9° were recorded; for the C2–C7 lordosis, a mean of 9.9° (SD ± 8.7°) and a median of 8.3° were recorded; for the C7 slope, a mean of 22.6° (SD ± 7.4°) and a median of 21.5° were recorded; and for the segmental Cobb angle, a mean of 7.3° (SD ± 7.4°) and a median of 6° were recorded (Table 3). Following a 12-month follow-up period, an additional set of radiographs was conducted to assess the selected cervical sagittal balance (CSB) parameters. During this evaluation, the parameters measured were as follows: for the C2–C7 SVA, a mean of 22.3 mm (SD ± 10.7 mm) and a median of 22.4 mm were recorded; for the SCA, a mean of 79.9° (SD ± 8.3°) and a median of 79.7° were recorded; for the C2–C7 lordosis, a mean of 10.9° (SD ± 8.7°) and a median of 9.6° were recorded; for the C7 slope, a mean of 20.3° (SD ± 7.2°) and a median of 19.7° were recorded; and for the segmental Cobb angle, a mean of 6.1° (SD ± 6.5°) and a median of 5.6° were recorded (Table 3). The alterations in the values of the examined parameters at the 12-month mark, compared to the values prior to the surgery, were as follows: for the ∆C2–C7 SVA, a mean of 5.8 mm (SD ± 5.7 mm) was recorded; for the ∆SCA, a mean of 7.1° (SD ± 5.1°) was recorded; for the ∆C2–C7 lordosis, a mean of 7° (SD ± 7.8°) was recorded; for the ∆C7 slope, a mean of 5.1° (SD ± 3.7°) was recorded; and for the ∆Cobb segment angle, a mean of 4.9° (SD ± 5.1°) was recorded (Table 3).

### 3.2. Clinical Outcomes

Prior to surgery, the preoperative assessment employing the VAS and NDI scales indicated mean scores of 5.9 (SD ± 2.3) and 23.8 (SD ± 8.78), respectively [points]. One month after postsurgery during the follow-up visit, these scores were reduced to 2.4 (SD ± 2.33) for the VAS and 14 (SD ± 7.7) for the NDI [points]. Upon completing the 12-month follow-up, the scores further decreased to 2.2 (SD ± 2.0) for the VAS and 10.9 (SD ± 8.7) for the NDI. Throughout the 12-month follow-up period, the absolute respective changes in the clinical parameters under study exhibited a mean of 3.7 (SD ± 2.7) points for the VAS and 13.1 (SD ± 9.9) points for the NDI. Our study also assessed the occurrence and relationship of the anticipated clinical endpoints with the sagittal balance parameters in the cervical region. A statistically significant difference was noted for the SCA in the groups with VAS < 1 and VAS ≥ 1, 84° vs. 79° (*p* = 0.0307), as well as for the alteration in the C7 slope values during the 12-month follow-up: 6.8° vs. 4.7° (*p* = 0.0453). Concerning the second assessed endpoint, i.e., for the NDI < 14 points, a significant statistical difference emerged for the C7 slope value after the 12-month follow-up, 19° vs. 22° (*p* = 0.0406), as well as for the segmental angle value after 12 months: 6.8° vs. 3.8° (*p* = 0.0417) (Table 4).

### 3.3. Subsidence

Out of the 95 patients evaluated, the phenomenon of subsidence was observed in 38 (40%) cases. The preoperative measurement of the sagittal balance parameters within this subset of patients revealed the following: for the C2–C7 SVA, a mean of 26.03 mm was recorded; for the SCA, a mean of 81° (SD± 9.8°) was recorded; for the C2–C7 lordosis, a mean of 9.5° (SD ± 10.7°) was recorded; for the C7 slope, a mean of 20.5° (SD ± 7.7°) was recorded; and for the segmental Cobb angle, a mean of 6.2° (SD ± 6.1°) was recorded. Among all the assessed parameters, only the C2–C7 SVA demonstrated a statistically significant difference between the groups with and without subsidence: 26.03 vs. 21.79 [mm], with *p* = 0.0182 preoperatively, and 27.80 vs. 24.94 [mm], with *p* = 0.0449 on the day after surgery, respectively (Table 5).

## 4. Discussion

Spinopelvic sagittal balance has been a widely studied concept since the 1990s [24,25,26,27]. While research has primarily concentrated on the thoracolumbar segment, where balance has been established through extensive studies of asymptomatic populations, the cervical region of the spine is gaining prominence. Due to its substantial mobility, the cervical segment primarily functions to compensate for changes occurring in the lower spinal sections [2,5]. However, pathological changes, which are most commonly degenerative in nature, can disrupt the inherent sagittal balance of the cervical segment to such an extent that it becomes incapable of fulfilling its function [4,28]. The surgical treatment undertaken often leads to spondylodesis between individual vertebral bodies in the cervical region, which, by reducing the number of mobile segments, impairs the ability to maintain its own balance and compensate for changes in spinopelvic balance. For this reason, it is important to think about the possible impact on CSB as early as the treatment planning stage. Numerous researchers have conducted investigations into the assessment of radiological parameters related to CSB. These studies have observed alterations in these parameters in cases of pathology and have explored their correlation with clinical indicators [4,7,9,29,30,31,32,33]. In a comprehensive literature review published in 2018 by Ling in collaboration with Le Huec, they highlighted the C7/T1 slope, cervical sagittal vertical axis (cSVA), and spinocranial angle (SCA) as pivotal parameters in assessing CSB [5]. These parameters have a significant impact on overall balance, with the SCA serving as a well-correlated indicator of C2–C7 lordosis [5,8]. The current study aimed to assess the temporal evolution and its impact on clinical parameters, as indicated by the VAS and NDI scales, of the aforementioned CSB parameters. Furthermore, alterations in these parameters were scrutinized in a cohort of patients who experienced implant subsidence, which were juxtaposed with a group where this phenomenon was absent. 

### 4.1. Cervical Sagittal Balance and Subsidence

The scientific literature lacks an abundance of investigations into the correlation between subsidence and CSB parameters. Notably, in 2017, Lee et al. published a study involving 41 patients, wherein they revealed that a T1 slope < 28° might constitute a risk factor for subsidence [19]. In our study, we did not observe a statistically significant difference in the C7 slope in the groups with and without subsidence at any evaluated time point (Table 4). Since C7 and T1 are considered as corresponding parameters, we conclude that our results do not reflect the findings of Lee et al. [2,5,19]. Furthermore, a statistically significant distinction between the subsidence and nonsubsidence groups was evident in the cSVA parameter, both preoperatively (26.03 vs. 21.79 mm, *p* = 0.0182) and postoperatively (27.80 vs. 24.94 mm, *p* = 0.0449), as revealed by multivariate analysis. The authors postulate that a higher cSVA might influence subsidence by unevenly distributing the forces exerted by the implant on the vertebral body endplates. Asymmetric pressure on the endplate could be prompted by an imbalanced SVA, thereby potentially contributing to subsidence. Nevertheless, drawing definitive conclusions is challenging due to the paucity of information on this topic in the existing literature. This finding, though applicable to the current study population, might lack statistical significance when extended to a broader cohort. The subsequent parameter displaying statistical relevance was the segmental angle (°), thereby showcasing a difference of 4.12° vs. 7.39° (*p* = 0.0144) in patients with and without subsidence, respectively, after the 12-month follow-up. The authors attribute this disparity to the reduction in segmental lordosis resulting from the implant’s collapse into the vertebral bodies. 

### 4.2. Cervical Sagittal Balance and Clinical Outcomes

Two clinical endpoints were assessed: VAS < 1 point and NDI ≤ 14 points, which were indicative of complete pain resolution and neck disability that does not impede regular functioning [22,23], respectively. The study aimed to determine if specific cervical sagittal balance parameters could predict higher scores in the VAS and deterioration in the NDI scores. In 2020, Zaidman et al. published a paper revealing a statistically significant negative correlation between C2–C7 lordosis (C2–C7 regarding the Cobb angle) and NDI scores, thereby indicating that lower lordosis corresponds to higher NDI scores. Other authors have also endeavored to explore the impact of cervical sagittal balance parameters on clinical outcomes [19,34]. Current knowledge indicates that alterations in cervical sagittal balance contribute to susceptibility to adjacent segment disease, thus consequently impacting treatment outcomes negatively [35,36]. In a 2012 study, Tang et al. demonstrated a positive correlation between the C2–C7 sagittal vertical axis (SVA) and the neck disability index (NDI) while finding a negative correlation with the SF-36 scores [37]. Building upon this, we hypothesize that C2–C7 SVA values influence clinical parameters and quality of life. However, our study did not yield results aligning with the aforementioned correlations, as both the visual analog scale (VAS) and NDI values of the C2–C7 SVA were comparable (Table 5). Nevertheless, a significant statistical difference in the spinocranial angle (SCA) values was observed in patients with VAS < 1 and ≥1, specifically for 84 vs. 79 degrees, respectively, with *p* = 0.0307. This implies that, in our study population, higher SCA values correlated with improved pain relief. It is worth noting that the SCA values in both groups were found to be within the established normal range of 83 ± 9 degrees [5,8]. Consequently, it is plausible to postulate that the spinocranial angle (SCA) plays a significant role in influencing pain relief, although the precise mechanisms remain elusive. Similarly, the pattern observed for the C7 slope is noteworthy; it exhibited a statistically significant disparity between the group with NDI ≤ 14 points and the group with NDI > 14 points, thus registering 19° versus 22°, respectively, with a *p* value of 0.0406. This implies that individuals with smaller C7 slope angles were associated with mild or minimal disability. Deciphering whether these selected parameters are inherent characteristics of the studied population or indeed exert tangible influence on the settlement and clinical outcomes of the procedure presents a challenge. Consequently, further research involving more extensive cohorts is imperative to refine and advance the realm of surgical intervention for spinal conditions.

### 4.3. Study Limitations and Prospectives

Certainly, our study is not without limitations, with the foremost among them being the modest sample size and the single-center nature of the investigation. Furthermore, the absence of established and universally accepted standards for sagittal balance parameters in the cervical spine poses a considerable challenge in precisely characterizing a balanced spinal configuration. This constraint is undeniably significant. However, the authors firmly believe that this work underscores the critical significance of contributing data pertaining to this matter. Consequently, it underscores the pressing need for intensified research endeavors aimed at enhancing the quality of care and ultimately ensuring the well-being of patients.

## 5. Conclusions

Attaining favorable clinical outcomes in anterior cervical discectomy and fusion (ACDF) treatments is a complex process influenced by multiple factors. Cervical sagittal balance parameters may play an important role in this issue. The present study indicates a possible influence of the C7 slope and spinocranial angle (SCA) on the clinical effects expressed in the NDI and VAS scales, respectively. Despite these insights, the authors advocate for a comprehensive approach, thus stressing the consideration of global sagittal balance over isolated cervical parameters during surgical planning. Regarding subsidence, higher pre- and postoperative sagittal vertical axis (SVA) values might impact its incidence rate. However, this phenomenon may depend on a variety of other factors such as the patient’s bone quality, implant type, and size or plating. The authors would like to underscore that while sagittal balance parameters are significant, they are part of a larger framework, which prompts a thorough evaluation of the full spectrum of risk factors available in the literature for treatment success.

## Figures and Tables

**Figure 1 biomedicines-11-03310-f001:**
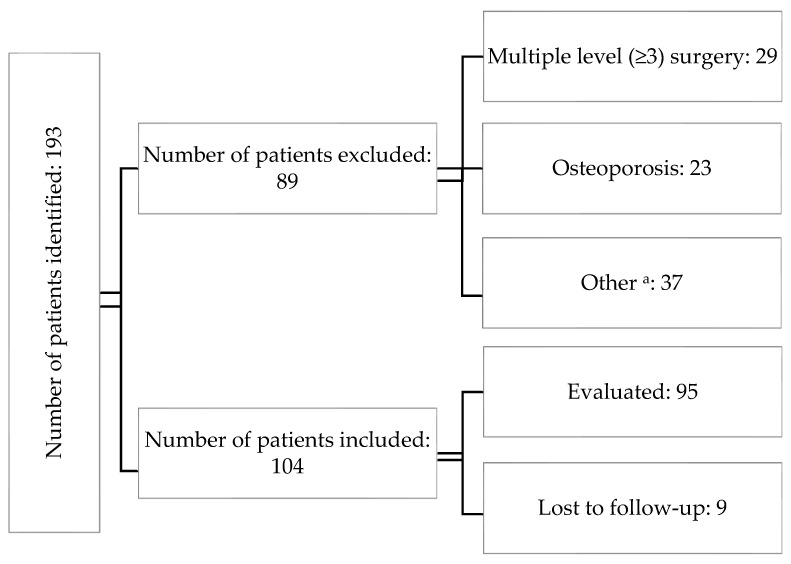
The chart introducing the patients’ flow throughout the study. ^a^—the “other” group encompassed patients who had active rheumatologic/metabolic diseases or had previously undergone surgery at a different level.

**Figure 2 biomedicines-11-03310-f002:**
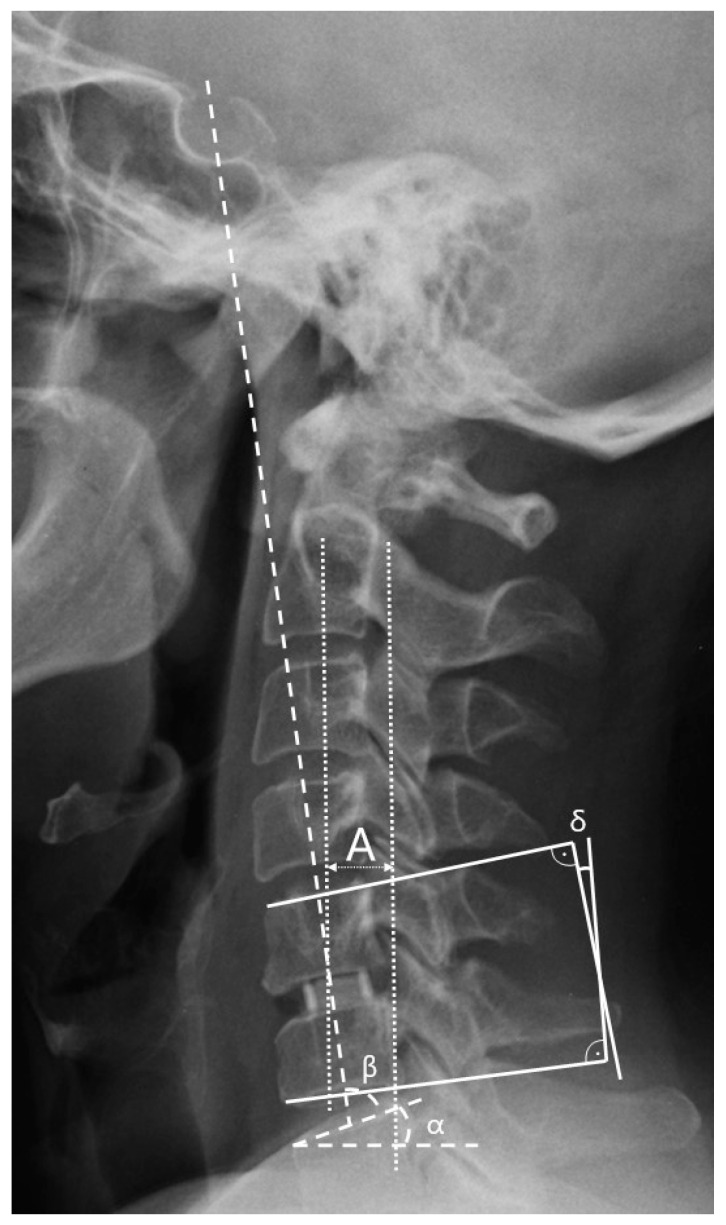
Measurement of individual selected parameters of cervical sagittal balance: A—C2–C7 sagittal vertical axis (C2–C7 SVA); α—C7 slope; β—spinocranial angle (SCA); δ—segmental (Cobb) angle.

**Figure 3 biomedicines-11-03310-f003:**
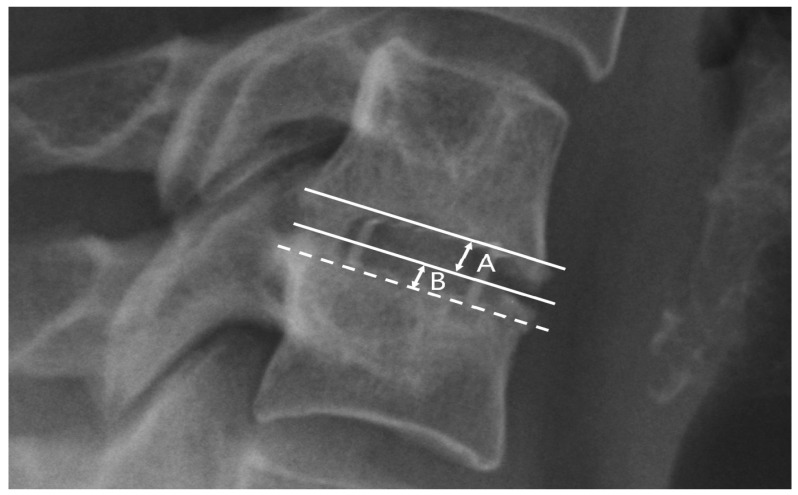
Assessment method for implant subsidence in fused vertebrae: B—subsidence depth; A—intervertebral height. Subsidence was defined as (B/A) ≥ 0.3.

**Table 1 biomedicines-11-03310-t001:** Characteristics of the study population (*n* = 95).

Characteristics	Value
Age, y, mean (range) (SD)	51, (31–73) (10, 24)
≥60 y, *n* (%)	19 (20%)
Gender: female, *n* (%)	67 (71%)
Type of spinal fusion:	
Single-level, *n* (%)	30 (32%)
Double-level, *n* (%)	65 (68%)
C3/C4, *n*	2 (2.1%)
C4/C5, *n*	0 (0%)
C5/C6, *n*	26 (28.4%)
C6/C7, *n*	2 (2.1%)
C3–C5, *n*	4 (4.2%)
C4–C6, *n*	15 (15.8%)
C5–C7, *n*	46 (48.4%)
Implant material:	
PEEK, *n* (%)	57 (60%)
TC-PEEK, *n* (%)	38 (40%)

**Table 2 biomedicines-11-03310-t002:** Cervical sagittal balance parameters selected for the study: definitions.

Parameter	Description
C2–C7 saggital vertical axis (cSVA)	The distance from the posterior superior corner of C7 to the plumbline from the centroid of C2.
Spinocranial angle (SCA)	The angle is measured as the deviation between the slope of C7 and the straight line that connects the midpoint of the C7 end plate to the midpoint of the sella turcica.
C7 slope	The angle between a horizontal line and the superior endplate of C7.
C2–C7 lordosis	The angle between the C2 and C7 lower endplates.
Segmental angle	The Cobb’s angle between the lower endplates of the fused vertebrae.

**Table 3 biomedicines-11-03310-t003:** Sagittal balance parameters—results of measurements in the assessed time points: ^a^—1 day before the surgery; ^b^—the day after the surgery; ^c^—after 12-month follow-up; and ^d^—average difference between the parameter value measured before the surgery and the value measured at the 12-month follow-up.

Parameter		Preoperative ^a^	Postoperative ^b^	After 12 mo Follow-Up ^c^	∆ pre-12 m ^d^
C2–C7 SVA [mm]	Mean:	23.5 (SD ± 11.5)	26 (SD ± 10.6)	22.3 (SD ± 10.7)	5.8 (SD ± 5.7)
	Median:	22	24.4	22.4	-
SCA [°]	Mean:	81 (SD ± 9.8)	79.5 (SD ± 7.3)	79.9 (SD ± 8.3)	7.1 (SD ± 5.1)
	Median:	79.8	79.9	79.7	-
C2–C7 lordosis [°]	Mean:	9.5 (SD ± 10.7)	9.9 (SD ± 8.7)	10.9 (SD ± 8.7)	7 (SD ± 7.8)
	Median:	8.6	8.3	9.6	-
C7 slope [°]	Mean:	20.5 (SD ± 7.7)	22.6 (SD ± 7.4)	20.3 (SD ± 7.2)	5.1 (SD ± 3.7)
	Median:	21	21.5	19.7	-
Segmental angle [°]	Mean:	6.2 (SD ± 6.1)	7.3 (SD ± 7.4)	6.1 (SD ± 6.5)	4.9 (SD ± 5.1)
	Median:	5.5	6	5.6	-

**Table 4 biomedicines-11-03310-t004:** Selected cervical spine balance parameter values based on clinical outcomes as assessed by VAS and NDI scales. ^a^—multivariate logistic regression model, including all cervical spine balance parameters at specific time points as quantitative variables.

			NDI ≤ 14 pts			VAS < 1 pts		
		N of Patients (%):	78 (82.1%)	17 (17.9%)		16 (16.8%)	79 (83.2%)	
			Yes	No	*p* Value ^a^	Yes	No	*p* Value ^a^
SVA 12 M [mm]	Mean:	12 M	22.6	21.9	0.3223	23	19.5	0.3204
		∆:	5.4	7.4	0.2112	6.0	5.7	0.4210
SCA 12 M [°]		12 M:	80.2	79.3	0.4809	**84**	**79**	**0.0307**
		∆	7.4	6.2	0.4777	8.6	6.9	0.3059
C2–C7 lordosis 12 M [°]		12 M:	10.33	10.8	0.7635	8.8	9.7	0.6678
		∆	5.8	7.7	0.7310	8.5	6.6	0.2221
C7 slope 12 M [°]		12 M:	**19**	**22**	**0.0406**	18.1	20.9	0.1339
		∆	5.4	4.1	0.0522	**6.8**	**4.7**	**0.0453**
Segmental angle 12 M [°]		12 M:	**6.8**	**3.8**	**0.0417**	4.9	5.7	0.5224
		∆	5.1	4.7	0.7832	5.5	4.6	0.5812

**Table 5 biomedicines-11-03310-t005:** Dependence of the occurrence of subsidence on selected cervical sagittal balance parameters over time: ^a^—Mann–Whitney U test; ^b^—*t*-test for independent variables; and ^c^—multivariate logistic regression model, including all cervical balance parameters in certain time point.

		Preoperative		Postoperative		After Follow-Up	
Parameter	Subsidence	Mean	*p* Value	Mean	*p* Value	Mean	*p* Value
C2–C7 SVA [mm]	Yes	26.03	**0.0478** ^a^	27.80	**0.0491** ^a^	23.09	0.4722 ^b^
	No	21.79	**0.0182** ^c^	24.94	**0.0449** ^c^	21.81	0.3499 ^c^
SCA [°]	Yes	81.39	0.7687 ^b^	79.38	0.8089 ^b^	79.95	0.9969 ^b^
	No	80.78	0.8476 ^c^	79.75	0.4435 ^c^	79.95	0.4758 ^c^
C2–C7 lordosis [°]	Yes	9.30	0.8197 ^a^	9.03	0.5818 ^a^	10.64	0.6134 ^a^
	No	9.69	0.5080 ^c^	10.49	0.8073 ^c^	11.05	0.9191 ^c^
C7 slope [°]	Yes	21.07	0.5388 ^b^	22.64	0.9589 ^b^	20.37	0.9565 ^a^
	No	20.07	0.5080 ^c^	22.56	0.7244 ^c^	20.29	0.7051 ^c^
Segmental (Cobb) angle [°]	Yes	5.82	0.8405 ^a^	6.02	**0.0406** ^a^	7.39	**0.0072** ^a^
	No	6.43	0.4745 ^c^	8.22	0.1738 ^c^	4.12	**0.0144** ^c^

## Data Availability

The datasets analyzed during the current study are not publicly available due to legal constraints but are available from the corresponding author upon reasonable request.

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
