# Peer review of "Cervical Sagittal Balance: Impact on Clinical Outcomes and Subsidence in Anterior Cervical Discectomy and Fusion"

_biomedicines, 2023, doi:10.3390/biomedicines11123310_

Round 1

Reviewer 1 Report

Comments and Suggestions for Authors

Dear Authors,

your paper seems very interesting and well structured.

Te quality o fpresentation is good, the methods are well explained and easy to be repeated.

Just some minor aspects have to be better addressed.

You should clearly define the study model. Is it an observational study on a single cohort without parallel group? Is it a case series? Please specify it.

With regard to the conclusion, maybe the absence of control should make you more cautious in the description of the importance of your findings. A brief revision/reorganization of the conclusion is suggested

Best regards

Author Response

Dear Reviewers & Editorial Team,

We would like to start by thanking you for all your time in evaluating our work submitted to your journal. We have read the reviewers' comments and made the appropriate corrections according to their content. Below is the content of the review with relevant commentary from us under each of the highlighted issues.

Reviewer:

Dear Authors,

your paper seems very interesting and well structured.

The quality of presentation is good, the methods are well explained and easy to be repeated.

Just some minor aspects have to be better addressed.

You should clearly define the study model. Is it an observational study on a single cohort without parallel group? Is it a case series? Please specify it.

Authors: Thank you for this suggestion! We have made the appropriate corrections to the manuscript. The study conducted was an observational study.

With regard to the conclusion, maybe the absence of control should make you more cautious in the description of the importance of your findings. A brief revision/reorganization of the conclusion is suggested.

Authors: Thank you for this suggestion! We have re-arranged the section under discussion so that the conclusions it draws are of a more 'cautious' nature and hook into the broader aspect of the problems presented in the paper.

Thank you once again for your comments and for taking the time to review this work. We hope that the corrections made will make it suitable for publication.

Sincerely,
Adam Bębenek M.D.

Bartosz Godlewski M.D., PhD

Reviewer 2 Report

Comments and Suggestions for Authors

I commend the authors on their remarkable research, titled "Cervical Sagittal Balance: Impact on Clinical Outcomes and Subsidence in Anterior Cervical Discectomy and Fusion." This retrospective study aims to evaluate how alterations in cervical spinal balance (CSB) parameters impact the clinical outcomes of the Anterior Cervical Discectomy and Fusion (ACDF) procedure, measured by VAS (Visual Analogue Scale) and NDI (Neck Disability Index), as well as the incidence of subsidence during a 12-month follow-up period. The subject matter explored in this study is compelling, and the manuscript exhibits clarity and readability. The introduction is well-crafted, the results are clearly presented, and the discussion is robust. The inclusion of radiographs depicting typical cases and measurements effectively enhances the overall comprehensibility of the manuscript.

However, certain critical aspects of this manuscript require further clarification before it can be considered suitable for publication. In the materials and methods section, it would be appropriate to include a flow chart presenting the reasons why 89 out of 193 patients were excluded. Additionally, in the same section, the authors stated that they incorporated two types of cages from the same manufacturer, but the name of the product, producer (company), city, and country of origin is not provided. The authors also failed to provide these data for the nanoparticle hydroxyapatite with which the cages were filled and the software used for statistical analysis. Moreover, the authors stated, "In order to categorize patients into distinct groups based on the cage material, a randomization table was employed." If so, how many patients were in each group? Were there any differences between the groups?

The conclusions section represents the weakest part of the manuscript. From a practical standpoint, how did the authors change the management of their patients? Do they recommend using Ti/PEEK or simple PEEK cages to their peers? Do they still favor stand-alone cages? Is attention to cervical sagittal balance parameters sufficient, or does the sagittal balance of the entire spine matter? Does bone quality, such as osteoporosis, play a role?

In summary, this manuscript shows promise; however, addressing the aforementioned points will undoubtedly enhance its contribution to the field and increase its suitability for publication.

Author Response

Dear Reviewers & Editorial Team,

We would like to start by thanking you for all your time in evaluating our work submitted to your journal. We have read the reviewers' comments and made the appropriate corrections according to their content. Below is the content of the review with relevant commentary from us under each of the highlighted issues.

Reviewer:

I commend the authors on their remarkable research, titled "Cervical Sagittal Balance: Impact on Clinical Outcomes and Subsidence in Anterior Cervical Discectomy and Fusion." This retrospective study aims to evaluate how alterations in cervical spinal balance (CSB) parameters impact the clinical outcomes of the Anterior Cervical Discectomy and Fusion (ACDF) procedure, measured by VAS (Visual Analogue Scale) and NDI (Neck Disability Index), as well as the incidence of subsidence during a 12-month follow-up period. The subject matter explored in this study is compelling, and the manuscript exhibits clarity and readability. The introduction is well-crafted, the results are clearly presented, and the discussion is robust. The inclusion of radiographs depicting typical cases and measurements effectively enhances the overall comprehensibility of the manuscript.

However, certain critical aspects of this manuscript require further clarification before it can be considered suitable for publication. In the materials and methods section, it would be appropriate to include a flow chart presenting the reasons why 89 out of 193 patients were excluded.

Authors: Thank you for this suggestion! We have inserted the corresponding flowchart into the manuscript along with its description. It is currently labelled as Figure 1.

Additionally, in the same section, the authors stated that they incorporated two types of cages from the same manufacturer, but the name of the product, producer (company), city, and country of origin is not provided. The authors also failed to provide these data for the nanoparticle hydroxyapatite with which the cages were filled and the software used for statistical analysis.

Authors: Thank you for this suggestion! We have added the relevant information on both implant and hydroxyapatite manufacturers to the manuscript. These can be found in the Materials and Methods section. A description of the statistical software used can also be found in the relevant part of the aforementioned section.

Moreover, the authors stated, "In order to categorize patients into distinct groups based on the cage material, a randomization table was employed." If so, how many patients were in each group? Were there any differences between the groups?

Authors: Thank you for this suggestion! Information on the number of patients with each implant has been added to Table 1. However, we decided to remove the fragment of the text in question (we mark it as crossed out in the manuscript) due to the fact that in its design it is not a study on the differences between individual implants. Especially since the same model of implants with the same shape and dimensions were used. Additionally, the authors have already published a comparative work on this matter, based on some of the material used in this work.

The conclusions section represents the weakest part of the manuscript. From a practical standpoint, how did the authors change the management of their patients? Do they recommend using Ti/PEEK or simple PEEK cages to their peers? Do they still favor stand-alone cages? Is attention to cervical sagittal balance parameters sufficient, or does the sagittal balance of the entire spine matter? Does bone quality, such as osteoporosis, play a role?

Authors: Thank you for these valuable comments! The conclusion section has been rearranged. We hope that the current content provides greater value and answers your questions. We again wish to address the issue of difference in implant coverage, recalling that this work is intended to focus on the importance of sagittal balance parameters. In the content of the section, we referred to the remaining risk factors, but we would like to note that in order to maintain a representative group of patients with "healthy" bone tissue in the population, we excluded patients with osteoporosis from the study, as it is a proven risk factor for subsidence. As for the issue of "stand alone cages", we refrain from making recommendations in this regard due to the fact that all our patients had stand alone cages and this study, due to its nature, does not allow conclusions in this regard.

Thank you once again for your comments and for taking the time to review this work. We hope that the corrections made will make it suitable for publication.

Sincerely,
Adam Bębenek M.D.

Bartosz Godlewski M.D., PhD

Round 2

Reviewer 2 Report

Comments and Suggestions for Authors

Well done.

Author Response

Thank you again for all your time! We appreciate your input in reviewing our article and improving it with your valuable tips.